# Recent Developments of Useful MALDI Matrices for the Mass Spectrometric Characterization of Lipids

**DOI:** 10.3390/biom8040173

**Published:** 2018-12-13

**Authors:** Jenny Leopold, Yulia Popkova, Kathrin M. Engel, Jürgen Schiller

**Affiliations:** Faculty of Medicine, Institute for Medical Physics and Biophysics, Härtelstr. 16/18, Leipzig University, D-04107 Leipzig, Germany; jenny.leopold@medizin.uni-leipzig.de (J.L.); yulia.popkova@medizin.uni-leipzig.de (Y.P.); kathrin.engel@medizin.uni-leipzig.de (K.M.E.)

**Keywords:** MALDI-TOF MS, MSI, matrix, lipid, phospholipid

## Abstract

Matrix-assisted laser desorption/ionization (MALDI) is one of the most successful “soft” ionization methods in the field of mass spectrometry and enables the analysis of a broad range of molecules, including lipids. Although the details of the ionization process are still unknown, the importance of the matrix is commonly accepted. Both, the development of and the search for useful matrices was, and still is, an empirical process, since properties like vacuum stability, high absorption at the laser wavelength, etc. have to be fulfilled by a compound to become a useful matrix. This review provides a survey of successfully used MALDI matrices for the lipid analyses of complex biological samples. The advantages and drawbacks of the established organic matrix molecules (cinnamic or benzoic acid derivatives), liquid crystalline matrices, and mixtures of common matrices will be discussed. Furthermore, we will deal with nanocrystalline matrices, which are most suitable to analyze small molecules, such as free fatty acids. It will be shown that the analysis of mixtures and the quantitative analysis of small molecules can be easily performed if the matrix is carefully selected. Finally, some basic principles of how useful matrix compounds can be “designed” de novo will be introduced.

## 1. Introduction

Next to spectroscopic techniques, such as infrared (IR) or nuclear magnetic resonance (NMR), mass spectrometry (MS) represents an indispensable method in the field of analytics and can be successfully applied in natural sciences and medicine [1]. Although the MS method was invented more than a hundred years ago when it was initially used by physicists to discover new elements [2], the application of MS in the life sciences and medicine is relatively new—in the earlier days of MS, electron-ionization-(EI)-coupled mass spectrometers were exclusively available [3]. Although EI is very useful to investigate small, volatile compounds which can be converted into (radical) ions by a collision with accelerated electrons, it regularly fails if large or non-volatile compounds, such as proteins or lipids have to be analyzed. The transfer into the gas phase ions without significant degradation and the need of derivatization step, prior to the analysis, is not as easy as it is for small molecules. Therefore, the invention of “soft ionization” methods—particularly electrospray ionization (ESI) and matrix-assisted laser desorption/ionization (MALDI)—that also enable the analysis of molecules which are refractive to EI, represented a definitive milestone in the history of MS [4]. Nowadays, “omics” approaches such as “proteomics”, “genomics”, and “metabolomics” are in the focus of research. “Lipidomics” is a part of the “metabolomics” field and was introduced and characterized for the first time by Fisher et al. [5], Han and Gross [6], and Spener and Lagarde’s research group [7,8], in 2003. Since then, the focus was on the (quantitative) analysis of the whole lipid composition and their alterations in biological samples, including the identification of lipid-mediated signaling processes in health and disease [6]. For further details about analytical approaches in the “lipidomics” field, please see, for example, the following references [9,10].

### 1.1. Soft Ionization Mass Spectrometric Methods

Nowadays, nearly all available ionization methods can be regarded as softer than EI [11]. Today, ESI and MALDI, the latter one most often combined with a time-of-flight (TOF) mass analyzer, are most commonly used, although ESI MS is currently more popular, due to the higher potential compatibility with different mass analyzers [12]. The significance of both soft ionization methods was emphasized by honoring the inventors of ESI and MALDI—Fenn and coworkers and Tanaka and coworkers, respectively—with the Nobel Prize for Chemistry in 2002 [13]. One major difference between MALDI and ESI is the sample introduction—the MALDI procedure occurs under a high vacuum (typically 1 × 10^−6^ mbar), while the ESI process occurs under ambient conditions. Since the focus of this review is on the MALDI-TOF MS, the fundamentals and applications of ESI MS will not be discussed here (for a general review, see for example Reference [14], and see Reference [15], especially, for lipid analyzation ). A detailed survey of the different typical applications of the MALDI-TOF MS and the most relevant methodological aspects are available in the book by Franz Hillenkamp and Jasna Peter-Katalinić [16], which is now already available as a second edition.

### 1.2. MALDI Mass Spectrometry

MALDI-TOF MS is based on the use of a matrix that absorbs the energy of the laser irradiation and is, thus, crucial for the successful generation of ions. There are basically three different classes of matrices available:Classical organic matrices, such as benzoic acid or cinnamic acid derivatives.Liquid crystalline matrices, which seem useful if a particular soft ionization is required.Inorganic matrices, such as graphite that provide only a weak background.

In this review we will focus on the first matrix class. The suitability of a compound as the MALDI matrix is dependent on the emission wavelength of the laser. Since the majority of MALDI studies were, so far, performed with ultraviolet (UV) lasers, we will focus nearly exclusively on this laser type, with an emission wavelength (λ) of either 337 nm or—more recently—355 nm. The first wavelength is typically emitted by an N_2_ laser, while 355 nm is characteristic of a neodymium-doped yttrium aluminum garnet (Nd:YAG) laser. During the last few years, N_2_ lasers are increasingly replaced by Nd:YAG since the N_2_ lasers have a shorter lifetime, compared to Nd:YAG lasers and, moreover, the photon energy at 355 nm is lower than 337 nm. This is beneficial if labile compounds have to be analyzed. Nevertheless, IR-MALDI MS is also quite useful and offers some important advantages which are summarized in Reference [17].

A crude scheme of the positive ionization processes taking place in a MALDI-TOF mass spectrometer is shown in Figure 1. When the pulsed laser beam hits the sample (normally co-crystals between the matrix and the analyte), its energy is primarily absorbed by the matrix that is present in an, approximately, thousand-fold excess over the analyte and has a significant absorption coefficient at the laser wavelength. Consequently, the matrix is vaporized, carrying the intact analyte molecules into the vapor phase. It has been suggested that MALDI is (among other reasons) such a soft process because the sample is expanded from the solid state into a high vacuum, leading to a significant cooling of the sample.

During the expansion process of the generated gas cloud, ions (e.g., H^+^ and Na^+^) are exchanged between the matrix and the analyte, leading to the formation of analyte ions. These analyte ions are called “adducts” or “quasi-molecular ions” because (in contrast to EI where radical cations are generated) they differ in their molecular weight, compared to the analyte. Along with cation generation (positive ion detection) anions (negative ion detection) can also be generated either by abstracting H^+^ or Na^+^ from the analyte or by fragmentation during the ionization process. Since all analytes with free electron pairs can be easily ionized by the attachment of cations, the positive ion mode is more popular. In contrast, the analyte usually needs to possess acidic functions (such as phosphate or carboxylate) in order to render negative ion detection possible. The ratio between the yield of cations and anions is determined by the (gas phase) acidities of the analyte and the matrix [16]. In contrast to the ESI MS, singly-charged, quasi-molecular ions are primarily generated upon the MALDI process. Therefore, the actually measured quantity, the mass-to-charge ratio (*m*/*z*), may be replaced directly by the monoisotopic mass of the analyte molecule, plus or minus the mass of the ion required to generate a charge. This makes the interpretation of the MALDI-TOF MS data, particularly simple.

After their generation, ions are accelerated in an electric field (typically about 20,000 V). After passing a charged grid, the ions drift freely over a field-free space (the drift zone) where the separation of the ions according to their *m*/*z* ratio is achieved—the lower the mass of the ions, the faster they arrive at the detector [18]. Peak resolution can be improved by using a reflector instead of the common linear TOF mass analyzer [16]. Therefore, reflector MALDI MS is frequently used for “lipidomics” studies.

## 2. The Role of the Matrix

Although there were considerable efforts to (i) understand the role of the MALDI matrix in detail and (ii) to design the most suitable matrix for a dedicated analytical problem, the finding of a useful matrix is usually a “trial and error” process. It is well-known that some samples can be easily analyzed in the presence of a dedicated matrix but the analysis fails if other, less suitable matrices or even derivatives of appropriate matrices are used. The application of a suitable matrix should result in (i) an excellent signal-to-noise (*S*/*N*) ratio, (ii) highly resolved spectra, (iii) negligible fragmentation of the analyte, and (iv) a moderate matrix background, in order to minimize interferences between the matrix and the analyte. The latter point is a particular problem for small molecules [19]. Although the finding of the optimum matrix is regularly still an empirical process, successful MALDI matrices must have different properties.

### 2.1. Requirements for the Suitability of a Compound as the MALDI Matrix

The MALDI matrices should fulfill the five following criteria:The matrix must possess a strong absorption at the emission wavelength of the laser—typically, in the UV range at either 337 or 355 nm. Therefore, all established organic matrices contain an aromatic ring system with delocalized π electrons, because the ionization efficiency (and, thus, the ion yield) increases with an increasing absorption coefficient of the matrix. This is the main reason why among the different dihydroxybenzoic acid (DHB) isomers only the 2,5-DHB isomer represents a useful MALDI matrix [20]. Although extinction coefficients are often determined in solution, it has to be emphasized that the absorption properties of MALDI matrices should be determined in the solid state, since only this approach mimics the conditions of the MALDI MS [20]. Since UV absorptions are generally rather broad, it is actually not a problem if the monochromatic laser light does not exactly fit to the UV maximum of the matrix. Therefore, optimized matrices for the 337 nm lasers may also be used at 355 nm.A useful matrix ensures the ion formation of the analyte. Since the often used aromatic ring system is poorly soluble in polar solvent systems, the carboxylic acid (normally benzoic acid or cinnamic acid derivatives) is an often used structure of matrix compounds. Since the carboxylic group is both polar and acidic, it ensures the solubility of the matrix in polar solvents [21] and the protonation of the analyte, respectively.The matrix should be stable under high vacuum conditions for as long as possible. Although this seems trivial, there are many promising MALDI matrices (e.g., 4-nitroaniline (PNA) or 2,6-dihydroxyacetophenone) which fail to fulfill this condition. Since the MALDI ionization process occurs under high vacuum conditions (normally about 1 × 10^−8^–1 × 10^−9^ bar) many compounds undergo a sublimation process. This leads to a continuous change of the matrix/analyte ratio and this may be one reason why the MALDI mass spectra show time-dependent changes.A perfect matrix should isolate the generated ions and prevent the generation of analyte clusters, for instance, dimer formation. Such clusters would complicate the spectra and decrease the sensitivity. Cluster avoidance is the primary reason why a significant excess of the matrix over the analyte should be used.The crystallization between the matrix and the analyte leads to co-crystals which should be as homogeneous as possible. The improvement of the crystal homogeneity is very important since this determines the “shot-to-shot” reproducibility of the acquired MALDI mass spectra. This important topic has been reviewed several times [22,23] and will be, thus, only loosely discussed here.

In sum, the simpler the sample preparation method, the more limited is the homogeneity of the matrix/analyte mixture. The most frequently used “dried droplet” method, i.e., the successive deposition of matrix and analyte, is regularly not suitable for quantitative analysis because it leads to inhomogeneous matrix/analyte co-crystals—particularly if different solvents with different volatilities for the matrix and the analytes are used. In the first step, the more volatile solvent evaporates and leaves the compound, which is more readily soluble in this solvent. The second compound crystallizes only when the less volatile solvent evaporates. Luckily, there are improved sample preparation methods (such as, electrospray deposition), which provide more homogeneous matrix/analyte mixtures. Many MS companies provide MALDI targets of different materials (aluminum, polished steel, etc.). However, these different metals only play a role if a very careful sample preparation is performed; using the common “dried droplet” method, the sample/matrix layer possesses a considerable thickness, so that the laser irradiance (that normally affects only the upper few micrometer of the sample) is not influenced by the metal at the bottom. This has been discussed in-depth in Reference [24].

### 2.2. Commonly Used MALDI Matrices

Although many different compounds have been discussed as potential MALDI matrices, only a handful of these are routinely used. Due to the limited space (for a more detailed review of the important MALDI matrices see Reference [25]), we will focus here on established organic molecules suitable for lipid analysis. All MALDI matrices which will be discussed in this review are shown in Figure 2. These compounds can be sorted into the following classes:Benzoic acid derivatives.Cinnamic acid derivatives.Heterocyclic matrices.“Other matrices”, for instance, acetophenone derivatives.

## 3. Which Matrix Fits Which Lipid Class?

Since MALDI is commonly used in protein and peptide research, the majority of the so far available reviews on the useful MALDI matrices are dedicated to polar molecules. In contrast, there is only a small number of reviews published about the analysis of lipids alone (e.g., Reference [26,27], or for a more general analyses, see Reference [10,12]). The most relevant lipid classes will be discussed here in the order of increasing molecular weights and the focus will be on the MALDI MS analysis of glycerolipids (particularly, triacylglycerols (TAG)) and glycerophospholipids, commonly named as phospholipids (PL).

### 3.1. Free Fatty Acids

Although fatty acyl residues are important constituents of all lipid classes (vide infra) and can be easily differentiated, due to their different molecular weights, the MALDI MS analysis of free fatty acids (FFA) is challenging if common organic matrices are used. The particular problem is the overlap between the signals of the FFA and the matrix-derived signals. Additionally, the high intensities of the matrix signals also reduce the sensitivity to detect FFA. These problems might be surprising, since all MALDI matrices possess a defined molecular weight and should, thus, provide a small number of peaks. DHB (molar mass (M) of 154.1 g/mol), for instance, should provide two peaks in the positive ion mode at *m*/*z* 155.1 and 177.1 for the H^+^ and Na^+^ adduct, respectively. Nevertheless, there are many more peaks which stem from the photochemical reactions of the DHB matrix, caused by the laser irradiation.

There are so far only a very few possibilities to overcome these phenomena. One possibility is the application of meso-tetrakis(pentafluorophenyl)porphyrin (MTPP) [28] as a matrix. MTPP is characterized by a considerable molar mass of M = 974.57 g/mol and does not undergo a gas phase fragmentation which would result in peaks below *m*/*z* 500. Therefore, FFA with M ≤ 350 g/mol, can be detected without a matrix overlap. This was illustrated by using a mixture of FFA derived from vegetable oils, subsequent to alkaline hydrolysis [29]. Although this approach works, it is not in use anymore because (i) this approach is rather insensitive and (ii) only saturated FFA can be identified, unambiguously. Unsaturated FFA always results in an additional peak with a mass difference of 14 Da. The assignment of this product is not clear but the mass difference would fit to the oxidation of a methylene into a carbonyl group. Surprisingly, the acquisition of the negative ion spectra was not possible under these conditions. In 2009 [30], it was shown that FFA are detectable as negative ions, if strong matrices with marked basicity are applied. In the presence of 1,8-bis(dimethylamino)naphthalene (DMAN), a “superbasic” matrix with a pKa of 12.21 [31], all investigated FFA could be detected as negative ions in the low picomole range. However, it was reported that DMAN is not stable under high vacuum conditions and, thus, its spectral quality changes with time [32]. Meanwhile, in another approach the matrix was also modified at the amino residues to reduce the volatility and to overcome the related problems [33].

Concomitantly, it was shown that the detection of FFA is more difficult the higher the double-bond content. This is caused by the limited vacuum stability of unsaturated FFA, such as arachidonic (20:4) and docosahexaenoic acid (22:6). In order to overcome this problem, Pirkl and coworkers [34] performed the analysis of FFA, under intermediate vacuum. This study is also remarkable because no standard MALDI matrix was used but the FFA were desorbed from insect wings. Although the detailed ionization mechanism is so far unknown, very “clean” spectra with high sensitivity could be obtained.

FFA can be also detected with 9-aminoacridine (9-AA), as a matrix. This compound is also characterized by alkaline properties (pKa 9.99) and provides only weak background signals [35]. With 9-AA, femtomolar amounts of FFA could be detected and the detector response was linear over two orders of magnitude of concentration. Therefore, 9-AA is nowadays a promising matrix in the field of quantitative “metabolomics” studies of negatively-charged compounds; its use is not limited to FFA [36]. Since FFA are of significant interest, there are quite often suggestions of new and promising matrix compounds. One auspicious new matrix for the MALDI-based analysis of FFA, investigated by Ibrahim and coworkers, is 1,6-diphenyl-1,3,5-hexatriene (DPH) [37], due to its negligible background signals under low laser fluences. In this context, the combination of DPH and elevated laser fluences allow the investigation of the fatty acyl composition of lipids, since this condition leads to the cleavage of the fatty acyl residues.

Nevertheless, the analysis of small molecules, such as FFA, is still difficult if classical organic matrices are used. Therefore, there were a lot of attempts to use inorganic matrices like graphite, which does not provide a significant background and is, therefore, a very useful matrix compound [38]. Unfortunately, there are indications that graphite leads to a considerable contamination of the ion source of the mass spectrometer and, thus, requires frequent cleaning, which is very time-consuming. Accordingly, there were also attempts to synthesize less volatile nanomaterials. For instance, it has been shown that the use of tellurium nanosheets is an excellent technique to analyze FFA and other small molecules [39]. A rather new matrix is represented by microalgae as an example of bio-nanostructures [40]. Although the focus of this paper is on “classical” organic (not inorganic matrices), Budimir and coworkers [41] have also demonstrated the application of desorption/ionization on porous silicon to monitor deprotonated FFA. Unfortunately, the sensitivity of this approach is quite poor. The determination of the double-bond positions in unsaturated fatty acids is still a challenging task and normally requires a selective cleavage at the double-bond position (for instance, by ozone [42] or the Paternò-Büchi reaction [43]), the derivatization of the generated carbonyl compounds, and their subsequent identification. It was recently shown that the charge-switch derivatization of free fatty acids with *N*-(4-aminomethylphenyl) pyridinium and subsequent tandem MS (either MALDI or ESI) allows the identification of fatty-acid double-bond positional isomers [44,45].

### 3.2. Cholesterol and Cholesteryl Esters

In addition to membrane proteins and glycerophospholipids, cholesterol is another important component of biological membranes. The recently introduced term “cholesterolomics” confirmed the importance of cholesterol and derivatives derived thereof [46]. Using common organic matrices such as DHB, cholesterol is not detectable as the expected H^+^ or Na^+^ adduct but only subsequent to water elimination at *m*/*z* = 369.3 ([M + H − H_2_O]^+^) [47]. Although it has been shown that cholesterol can be detected by MALDI-TOF MS [48], due to the relatively small molar mass of cholesterol, there is a considerable overlap with the matrix background—as already discussed for the FFA. As far as we can say, there is only a limited interest in the MALDI-TOF MS determination of cholesterol since such metabolites can be easily determined either by liquid chromatography (LC)-coupled to MS from different biological samples [49,50] or by simple and relatively inexpensive, commercially available enzymatic kits [51].

Since the hydroxyl group is lacking in cholesteryl esters (CE), the H_2_O elimination is not a problem. Although CE occur in large amounts in lipoproteins and are of considerable medical relevance, there were yet only few MALDI MS studies [48,52] at which DHB was used as the MALDI matrix. In a recent study, a MALDI-based method was developed to examine the most abundant constituents of lipoproteins [53]. Subsequent to the separation of different lipid classes, between 150 and 200 lipids were detected in the different lipoprotein fractions. A similar study was performed by Holčapek and coworkers [54], who investigated the polar lipids by the ESI MS. However, the apolar lipid constituents (sterols, CE, TAG and diacylglycerols (DAG)) were determined by the MALDI-TOF MS, since these compounds provide an improved ion yields, under these conditions. In 2011, Zaima et al. identified two potential CE biomarkers (cholesterol linoleate (CE 18:2) and cholesterol oleate (CE 18:1)) in mouse and human lipid-rich regions of atherosclerotic lesions, with imaging mass spectrometry (MSI), using DHB as matrix [55]. Similar to DAG and TAG (see below), CE are exclusively detectable as alkali metal adducts and not as H^+^ adducts. Although this aspect was not yet investigated in detail, it is reasonable to assume that a similar mechanism, as in the case of the TAG, applies [56]—protonated CE are not stable enough to “survive” the flight distance (TOF) without pronounced fragmentation, compared to the Na^+^ adducts. This agrees with the observation that even the MALDI mass spectra of chromatographically-purified CE preparations do always contain free cholesterol. However, it was shown that the conversion of the CE into free cholesterol can be attenuated if the sodium salt of the DHB, instead of the free acid, is used [57].

### 3.3. Glycerolipids and Glycerophospholipids

Selected lipids are of significant diagnostic interest. This applies, for instance, for atherosclerosis [58] and rheumatic diseases [59]. One important reason is that some (phospho)lipids, such as lysolipids and phosphatidic acid (PA), for example, play an important physiological role because they represent (lipid) second messengers [60]. A detailed survey of lipids and their biological significance is available in [61].

#### 3.3.1. Di- and Triacylglycerols

TAG are important nutritional sources of energy (e.g., vegetable oils) and are essential to store excessive energy (adipose tissues in living organisms), while DAG are relevant as second lipid messengers [62]. The positive ion MALDI-TOF mass spectra of both DAG [63] and TAG [64] yield exclusively the corresponding Na^+^ adducts. Additionally, both lipid classes result also in considerable yields of fragment ions, which can be explained by the loss of one sodiated FFA [65]. High energy collision-induced dissociation (CID) experiments can be, thus, used for the structural elucidation of TAG [66] and the identification of the two fatty acyl residues. The MALDI characteristics of the DAG and TAG are largely independent of the applied matrix [63,64]. In an older study, Asbury and coworkers found that the achievable sensitivities for individual TAG species could be influenced by the solvent system—DHB and α-cyano-4-hydroxycinnamic acid (HCCA) both dissolved in acetonitrile/H_2_O + 0.25% trifluoroacetic acid (50/50), as well as dithranol in chloroform (first reported as the new MALDI matrix by Juhasz and Costello [67]) result in sensitivities in the picomole range [64]. In contrast, a much higher sensitivity could be achieved if the solvent was altered, as DHB in acetone gave a detection limit in the femtomole range [64]. However, sensitivity is usually not a major issue because TAG are available in high amounts in lipid-rich biological samples. In the positive ionization mode, DHB provides a less pronounced background than the cinnamic acid derivatives [68] that tend to generate matrix ion clusters. It is a major advantage of the MALDI-TOF MS that even higher salt contents of the sample are tolerated, which opens the possibility to add dedicated salts as dopants [69]. In a 2008 study it was shown that 9-AA, as another suitable matrix in the positive ionization mode, allows the detection of TAG, only in the presence, but not absence, of dopants such as ammonium acetate [70]. Lou et al. demonstrated in 2015, that HCCA dissolved in tetrahydrofuran can be used to increase the detectability of TAG in the presence of PL and reduce the known ion suppression effect of PL in multicomponent mixtures [71].

It is remarkable that both, DAG and TAG (as well as CE), appear exclusively as alkali metal ions, whereas the proton adducts are never detected, even if acidified solutions are investigated. A convincing explanation has been given by experiments where the yield of fragment ions (stemming from the loss of one acyl residue) could be significantly reduced at alkaline conditions [56]. With this in mind, it was suggested that the observed fragments actually arise from unseen protonated TAG because their complete fragmentation occurs so rapidly that protonated TAG are not observed. If the pH is increased, the H^+^ concentration is simultaneously decreased leading to a reduced H^+^ adduct generation and, accordingly, to a lower yield of fragmentation products [56]. Despite this limitation, it has been repeatedly shown that MALDI-TOF MS is suitable for the screening of the compositions of crude TAG mixtures, for instance vegetable oils [72], to monitor the adulteration of more expensive oils by cheaper oils, such as corn oil [73], and even to monitor the crystallization behavior of milk fats which depends on the TAG composition [74].

Further applications range from pathophysiological aspects [75,76] over nutrition- related topics [77,78] to textile surfaces analysis [79] and forensic studies [80]. The most frequently used matrix for the TAG analysis was DHB, as well, either as a solution in methanol/H_2_O (7/3, *v*/*v*) [79] or in acetone/H_2_O (1:1 *v*/*v*, supplemented with 0.1% trifluoroacetic acid) [80], or applied by sublimation [77,78]. Furthermore, nanoparticles from materials, such as silver salts [76,81] or graphite [82], have been introduced as novel matrices. However, detailed information would be outside of the scope of this review.

#### 3.3.2. Phospholipids

Since there is currently a significant interest in “lipidomics”, many different methods of lipid analysis have been developed. In addition to MS there are also methods based on chromatography and spectroscopy (for instance, NMR). Nevertheless, MS is probably the most powerful method to elucidate the lipid composition of an unknown sample, due to its high sensitivity [83,84,85]. Some selected PL which occur in the majority of tissues and body fluids are shown in Figure 3. Please note that all these compounds differ in their charge states. Therefore, the mass spectrometric detectability of the different PL, differs significantly. This aspect will be outlined below, in more detail, because this hinders the analysis of lipid mixtures significantly.

As already indicated above, MS represents the most powerful tool to investigate PL and all modern soft ionization techniques have already been successfully applied to PL, although ESI [86] was so far only primarily used in “lipidomics” studies [87]. However, the number of MALDI-TOF MS-based lipid studies is increasing [12]. The reason for this is the corresponding imaging approach (MSI) which attracts significant interest, as it enables the spatially-resolved determination of lipids in biological tissues [88]. As all tissues are composed of cells, which have considerable amounts of PL in their membranes, lipids are easily-detected by MSI experiments. In the last decades, the research interest has shifted from “proteomics” to “lipidomics”, since lipids are more easily detectable by MSI. Another advantage is that lipids are ubiquitous compounds, which appear in all vertebrates, to a similar extent [89].

Despite the considerable progress in the MALDI-TOF MS hardware, the question for the “optimum” matrix has not yet been answered. It seems likely that the matrix of choice depends on the scientific question and the investigated lipid species.

The importance of the matrix on the quality of the MALDI mass spectra of PL was already outlined in 1995, in a pioneering study by Harvey [90]. Although many different matrix compounds, such as 1,4-dihydroxy-2-naphthoic acid, 3,7-dihydroxy-2-naphthoic acid, 6,7-dihydroxycoumarin (aesculetin), 7,8-dihydroxy-6-methoxycoumarin, 4-hydroxycoumarin, 7-hydroxycoumarin, 7-hydroxycoumarin-4-acetic acid, 3-hydroxypicolinic acid, and 2,5-dihydroxyterephthalic acid were tested, sinapinic acid, HCCA, and DHB gave the optimum results regarding the achievable sensitivity, resolution, and the extent of observed fragmentation. To these authors’ opinion, DHB is the matrix of choice and all studies should be initiated by using DHB. Independent of the used matrix, one should strictly keep in mind that the individual PL classes are detected with highly different sensitivities [91]—phosphatidylcholine (PC) is most sensitively detected as a positive ion (detection limit about 0.5 pmol or 0.4 ng PC), while significantly higher amounts of, for instance, phosphatidylethanolamine (PE) are required to obtain a comparable *S*/*N* ratio. This is caused by the permanent positive charge of the quaternary ammonium group of the PC molecule (Figure 3), while the similar functional group of the PE (hydrogens instead of methyl groups) can be easily deprotonated, leaving the negative charge of the phosphate group [92]. Expectedly, phosphatidylinositol (PI), phosphatidylserine (PS), and PA are detectable, with even lower sensitivities, due to their negative charge, i.e., their enhanced pK values, compared to PC [91]. This result could be confirmed and extended in later studies [93]. It was also recognized that the different detectabilities of the individual PL make the analysis of mixtures difficult—equally, if positive or negative ion detection is used. Therefore, the addition of at least one internal standard per lipid class is necessary for the quantitative analysis. Otherwise, the relative data, where just the intensities of dedicated peaks are compared, could be used, too. Of course, relative data have the disadvantage that they show only differences from one spectrum to another. To monitor changes in intensity levels from one or more compounds, absolute data are required.

Even though MALDI tolerates salt impurities to a significant extent, measurements without careful sample desalting result in complicated mass spectra, since each PL species shows several signals caused by the simultaneous generation of H^+^ (from the matrix) and alkali metal adducts (from sample impurities), making peak assignments ambiguous. For instance, the H^+^ adduct of 1-palmitoyl-2-arachidonoyl-*sn*-phosphatidylcholine (PC 16:0/20:4) and the Na^+^ adduct of 1-palmitoyl-2-oleoyl-*sn*-phosphatidylcholine (PC 16:0/18:1) cannot be differentiated if the resolution of the used mass spectrometer is not sufficiently high.

There are three possibilities to overcome this problem: (i) The careful desalting of the sample, (ii) the separation, on a chromatographic column, to isolate the different lipid species, or (iii) the addition of a large amount of auxiliary salts, such as CsCl [94]. The addition of Cs^+^ leads to a sufficiently high mass shift to guarantee that all the observed mass differences are caused by differences in the fatty acyl compositions of the different lipid species. This is sometimes a suitable alternative to tandem MS (MS/MS) experiments, in order to identify the fatty acyl composition. It was also shown that the presence of CsCl helps to detect small amounts of PL, in the presence of an excess of TAG, i.e., if extracts of adipose tissues are investigated [95].

As indicated above, PC suppresses the detection of other PL species in the positive ion mode. Therefore, it is reasonable to assume that the negative ion mode would be the optimum approach to detect all other PL, in the presence of major amounts of PC. However, there is evidence that PC also affects the negative ion data, depending on the applied matrix [96] (Figure 4).

Figure 4 provides clear evidence that the application of DHB and 9-AA result in characteristic differences when PC 16:0/18:1 (M = 759.6 g/mol) and 1-palmitoyl-2-oleoyl-*sn*-phosphatidylethanolamine (PE 16:0/18:1; M = 717.5 g/mol) were investigated. There are only minor differences in the positive ion spectra independent of whether DHB or 9-AA is used (Figure 4a), although the ratio between the H^+^ and Na^+^ adducts of PC 16:0/18:1 (e.g., *m*/*z* 760.6 (H^+^-adduct) and 782.6 (Na^+^ adduct) are different—obviously, 9-AA favors the generation of the H^+^ adducts, whereas the Na^+^ adduct is generated to a lesser extent. An explanation for this could be that 9-AA has a high affinity to sodium ions, corresponding to a higher binding affinity. Thus, the sodium-ion content would be reduced and less Na^+^ would be available to cationize the PC. Anyway, the negative ion spectra differ significantly depending on the applied matrix. For instance, just one signal is obtained if PE 16:0/18:1 is investigated in the presence of 9-AA (4b, bottom right), while many different signals (primarily stemming from the DHB matrix [68]) are observed in the presence of the DHB (4b, top right). Differences are even more obvious if the negative ion spectra of PC 16:0/18:1 are compared. The peak at *m*/*z* 744.6 which is seen in the presence of 9-AA reflects the loss of one methyl group from the PC headgroup (4b, bottom left). This methyl group loss is characteristic of 9-AA and does not occur in the presence of DHB, where a cluster ion between DHB and PC 16:0/18:1 is generated (4b, top left) at *m*/*z* 912.6. Therefore, PC may be misinterpreted as PE (and vice versa) if the negative ion spectra of lipid mixtures are recorded without the evaluation of the MS/MS fragmentations. This is explained in more detail in Reference [96]. It has been recently demonstrated that this problem can be completely overcome by the application of a binary matrix of DHB and 2,5-dihydroxyacetophenone [97]. Already these simple examples confirm the relevance of the matrix and that the correct choice of the most suitable matrix is extremely important.

### 3.4. Phospholipids: Some Selected Examples

#### 3.4.1. Oxidized Phospholipids

The choice of the optimum matrix is of particular importance for the detection of oxidation products of lipids. The “oxylipidomics” field is currently attracting increasing interest. Usually, the related analyses are performed by ESI MS [98] because this technique is somewhat softer than the MALDI and, thus, the detection of labile species (such as, hydroperoxides) is improved [99]. Lipid peroxidation products and oxidation products generated by hypochlorous acid (HOCl), an important oxidant generated, particularly, under inflammatory conditions [100], which adds to the unsaturated fatty acyl residues of lipids (leading to the generation of chlorohydrins and products derived, thereof [101]) are readily detectable in the presence of DHB [102], as well as other common matrices. The detection of other oxidation/chlorination products is a more challenging task. Oxidative modifications of the headgroups in chlorinated PE species (i.e., chloramines and dichloramines) are not detectable at all by the MALDI-TOF MS (in contrast to ESI MS) [103], under the DHB application. These chloramines are only detectable with 4-chloro-α-cyanocinnamic acid (ClCCA, a rather new chlorinated analogue of cinnamic acid) as a matrix [104] (Figure 5), although the molecular reasons for this discrepancy are not yet understood. A recent review dealing with such aspects is available in Reference [105].

Potential mechanisms which might explain these surprising differences are probably related to the different proton affinities of the matrices [106]. Such reasons may also explain the poor detectability of the typical lipid oxidation products, such as peroxides, which were so far detected by the MALDI-TOF MS only with a very low sensitivity or under harsh oxidation conditions, which were far from the physiological conditions [107]. A comprehensive review dealing with the detection of oxidized lipids by different MS methods appeared a few years ago [89]. In general, the detection of oxidation products of lipids by the MALDI-TOF MS has to be improved. This particularly applies in the context of MSI, where studies according to the detection of oxidized lipids are, so far, missing.

#### 3.4.2. Phosphorylated Phosphatidylinositols

There are different lipid classes with high biological significance but poor detectability. This applies, for instance, for (poly)phosphorylated PI (PIP, see Figure 3) and is aggravated by the fact that the in vivo concentration decreases from PIP to bisphosphates (generally known as PIP_2_) and particularly triphosphates (PIP_3_). All individual PIP generate excellent signals in the negative ionization mode and can be detected down to about 1 nmol. Nevertheless, their detection thresholds are significantly elevated, compared to common PL. While PC are detectable (as positive ions) in amounts of about 3 pmol, nearly the three-fold amount of PIP_3_ is required to obtain a reasonable *S*/*N* ratio [108,109]. Although absolute amounts should not be taken very seriously, because they depend on the condition of the used mass spectrometer, it should be noted that the negative and the positive ion mode interact with each other. In the presence of other lipids, such as PC, the detection limits of PIP are increase although PC, as such, is not detectable in the negative-ion mode [110], but only as a cluster ion, with one deprotonated DHB ion (vide supra) [110].

#### 3.4.3. Cardiolipins and Phosphatidic Acids

Similar data as in the case of PIP were reported regarding the analysis of cardiolipins (CL) [111], which are characterized by a high number of charges as well as an elevated molecular weight. Both aspects lead to a decreased detectability of CL. However, comprehensive investigations on the detectability of PIP and CL by MALDI-TOF MS, are not yet available. Changes of the pH values might be suitable tools to compensate for different charges and, thus, to enhance the detection of otherwise less sensitively-detectable compounds. Unfortunately, this approach is limited since PL are known to hydrolyze under acidic and particularly alkaline conditions. This is extremely pronounced if the plasmalogen species (alkenyl-acyl-PL) are of interest. Traces of acids are already sufficient to hydrolyze the vinyl ether linkages of plasmalogens [112,113]. The detection of PA and its lyso-products, LPA, are also challenging, due to their charge and their small contribution to the biological systems. Tanaka and coworkers suggested the use of an inorganic zinc complex as a matrix [114], to selectively trap the LPA and, in this manner, to improve its detectability. This approach has been improved and is now also suitable to monitor sphingosine-1-phosphate [115].

To these authors’ best knowledge, DHB is the most “universal” matrix for lipid analysis because it is able to detect virtually all lipid species ranging from apolar (such as TAG) to very polar (e.g., PIP) lipids. However, there are two disadvantages: (i) DHB is a poor matrix for negative ionization, since it provides a significant background caused by characteristic matrix oligomers and (ii) DHB is not very sensitive, compared to other matrices. Using particular clathrate nanostructures, lysophosphatidylcholines (LPC) were detectable down to about 700 ymol (700 × 10^−24^ moles), i.e., a detectable signal could be achieved from the already 10 analyte molecules [116]. This sheds light on the enormous sensitivities of modern MS detectors. The application of non-conventional matrices is a developing field and seems to be particularly relevant to the field of MSI studies [117].

### 3.5. Glycolipids

The group of glycolipids is comprised of glycolipids, glycoglycerolipids, and glycosphingolipids (GSL) that share some common features—they all contain at least one sugar moiety that is glycosidically-linked to the lipid moiety, one or two fatty acyl residues and no phosphorus. GSL are cell surface lipids that are important for innate and adaptive immunity (very recently reviewed in [118], interact with signaling molecules, and are associated with many diseases, such as cancer [119] and infectious diseases (very recently reviewed in [120,121]). They are subdivided into cerebrosides and gangliosides. In contrast to cerebrosides, which only contain one monosaccharide moiety, galactosides are more complex sphingolipids, with more than one monosaccharide moiety.

In the literature there are already reports that have used the MALDI-TOF MS or MSI to investigate glycolipids in biological samples. Itonori and coworkers analyzed the differences in purified GSL, from silkworm larvae and pupae, using the MALDI-TOF MS with HCCA as matrix [122]. Caughlin and coworkers published a sublimation approach with 1,5-diaminonaphthalene (1,5-DAN) as a matrix to visualize gangliosides in rat brain [123]. Results from our own unpublished work led to the assumption that the interpretation of lipid mixtures containing GSL is misleading if 9-AA is used as matrix, because neutral GSL cannot be detected with 9-AA, probably due to the missing charge of the target molecules. In contrast to that the DHB results in suitable spectra where the GSL can be detected as Na^+^ adducts. 9-AA is the matrix of choice when it comes to the detection of acidic GSL, such as sulfoglycolipids, as negative ions [124,125,126]. Furthermore, there is a review from Meisen et al. that describes the different methods to detect GSL. In this paper it is emphasized that especially the structural GSL analysis can be improved by a combination of thin-layer chromatography (TLC), immunostaining of the TLC plate, and MALDI MS [127].

### 3.6. Problems Related to Mixture Analysis

As outlined above, there is one major problem related to the MALDI-based mixture analysis. Depending on the charge state and—to a minor extent—the molecular weight, different compounds are detected with different sensitivities, and this basically applies independent of the used matrix. Thus, sensitively-detectable compounds may suppress the detection of less sensitively-detectable ones. However, this ion suppression problem applies for all soft ionization MS techniques. This implies, for instance, that the positive ion mode is usually dominated by PC, due to the permanent positive charge of the quaternary ammonium group, while the negative ion mode is dominated by lipids containing sulfate residues (which are often present in glycolipids), i.e., strong electrolytes which are always deprotonated and, thus, negatively charged. Since each investigated lipid mixture is different, the knowledge about the detection limits of the different PL classes is of crucial importance. However, such data are rather scarce and so far there are only data for positive ion detection [91].

If the lipid compositions of the extracts of the body fluids (such as plasma or blood), cell suspensions, or tissues are to be analyzed by the MALDI MS, it is advisable to compare both the positive and the negative ion spectra. It is also useful to apply alkaline (e.g., 9-AA, pK 9.99) and acidic (e.g., DHB, pK 2.94) matrices. If the DHB is used, PC and other PL, such as LPC or sphingomyelin are detectable and the spectra also exhibit peaks of TAG, free cholesterol, and CE. In contrast, negatively charged PL, such PI, PS, PA, CL, as well as PE (which is readily deprotonated) are only detected (if at all) with low sensitivity, but appear with reasonable intensities in the obtained mass spectra, after a negative ionization. Nevertheless, the presence of salts may significantly influence the detection levels of other lipids [128]. The detection limits are also influenced by the age of the used mass spectrometer, due to an aging of the detector which negatively affects the detection limits. The most serious problem is the detection of PE in mixtures, with comparable amounts of PC. PE is severely suppressed in the presence of PC in the positive ion mode (vide supra), and, moreover, also in the negative ionization mode if the “wrong” matrix is used. This may be caused by the “in-source” loss of one methyl group from the PC species which renders them detectable as negative ions, raising signals in the same *m*/*z* range as the negatively charged PE. Two different approaches can help to overcome these detection problems.

#### 3.6.1. Separation of the Mixture into the Individual Lipid Classes

Subsequent to separation, all PL classes are easily detectable in the obtained fractions independent of negative or positive ion detection. This separation can be performed by high-performance liquid chromatography (HPLC) or TLC that is still widely used in lipid research [129]. Since ESI requires a liquid sample, HPLC is the separation technique of choice for the ESI MS applications. A combination of HPLC with MALDI is more difficult since MALDI requires a solid sample (co-crystals between sample and matrix). Even though there are commercial systems available, which enable the solubilization of TLC-separated lipid classes from the corresponding TLC plate and its direct infusion into the (ESI) mass spectrometer, TLC may be more easily-coupled to the MALDI-TOF MS, i.e., the entire plate is coated with a matrix and, afterwards, directly scanned by the laser, in the same way as done in MSI. This was initially established for IR-lasers with glycerol as a matrix [111] and nearly simultaneously for the UV-lasers and the DHB as the matrix [130]. Further aspects of applications of the TLC-MALDI-TOF MS are available in Reference [131]. The importance of the TLC-coupled to (MALDI-TOF) the MS, led to the establishment of the “MS grade” TLC plates, that are characterized by a superior purity and a reduced silica layer thickness (100 instead of 200 µm). This adaptation results in an improved sensitivity and mass spectra of higher quality [132]. Although TLC-MALDI-TOF mass spectra are not quantitative, so far, it was explicitly shown that the detection limits achieved by MS are much lower, in comparison to typical TLC staining protocols [133].

Since the removal of PC is normally the most important step to detect the residual lipid classes, a fast and easy to perform this method has been suggested [134]—a mini-column filled with a cation exchanger was used to adsorb the lipids with positive partial charges (such as PC) from the chloroform phase of brain extracts. Since PC was removed under these conditions, its suppression effect did not play any role and many other lipid classes became detectable as positive ions. In the negative ion mode, the achieved improved sensitivity led to the detection of even very rare lipids such as PIP_2_. This work has, thus, demonstrated the importance of PC removal [134].

#### 3.6.2. Choosing the Most Suitable Matrix

PE is normally a very challenging lipid class—it is suppressed by other, more acidic lipids (such as PI) in the negative ion mode, and by PC in the positive ion mode. An excess of PC over PE applies for nearly all biological samples. Some matrices were already suggested to overcome the suppression of PE, especially when MSI experiments were performed where no previous separation is possible. One of the first matrices was PNA [135] dissolved in chloroform/methanol (2:1, *v*/*v*). With this system it was possible to obtain many different PE species as negative ions without interferences of PC. This is exemplarily shown in Figure 6.

Unfortunately, PNA tends to sublime under high vacuum conditions in the mass spectrometer. Thus, the analyte/matrix ratio changes with time and this leads to significant changes of the MALDI mass spectra. 2-mercaptobenzothiazole and 2-(2-aminoethylamino)-5-nitropyridine were suggested as more powerful, vacuum stable matrices with similar properties as PNA [135,137]. Nowadays, 9-AA [70] is the matrix of choice to detect PE in the negative ionization mode, according to the following significant advantages:The identification of PE (even in complex mixtures) is possible as described above.Although 9-AA is an excellent matrix in the negative ionization mode, it is also useful for positive ionization and offers an even higher sensitivity than DHB.9-AA provides nearly, exclusively, the H^+^ adducts but only less intense peaks of Na^+^ adducts of the PL. This cannot be explained by H^+^ donating properties, since 9-AA has no acidic properties but may be explained by a considerable Na^+^ affinity of 9-AA.Many extracts of biological materials are contaminated with compounds, such as detergents or plasticizers. These compounds, however, are not detected in the presence of 9-AA because they are lacking charged groups, which is an obvious requirement of using 9-AA. In the same way, compounds such as DAG, TAG, and also glycolipids often need an additive, such as sodium acetate [53] for their visualization with 9-AA as matrix.

The most serious disadvantage of 9-AA is that its performance depends significantly on the solvent composition—isopropanol/acetonitrile (60:40, *v*/*v*) is the solvent of choice and there is a considerable loss of sensitivity if other solvent or solvent mixtures are used [70].

Along with the “typical” organic matrices, some particular compounds were also suggested as useful matrices for the PL analysis. For example, ionic liquids, such as salts between aliphatic organic [138] or aromatic amines [139] and HCCA, were introduced as useful alternatives. These compounds normally do not lead to elevated sensitivities but are often characterized by a smaller spot size. This improves sample homogeneity and, thus, variations from shot-to-shot can be minimized. Reduced analyte/matrix cluster formation is another advantage of this system [140]. Finally, PL seem to be stabilized if ionic liquid matrices are used [141] and this is accompanied by reduced fragmentation. Despite these advantages, ionic liquid matrices are not widely used. Further studies are needed to explore the capabilities of these compounds, when used as MALDI matrices.

### 3.7. MALDI Imaging

Developed in the early 2000s [142], MALDI Imaging (MSI) is quite a new method which is attracting increasing interest. Since there are many timely reviews [143,144], however, this aspect will be only shortly discussed here, with a focus on the suitable matrix compounds.

All the spectra shown in this review so far were recorded from lipid extracts, i.e., the samples were first homogenized and afterwards extracted. The extraction process provides many advantages; for instance, water-soluble compounds, such as salts, sugars or proteins can be readily removed and lipids can be easily enriched by the evaporation of the organic solvent. However, the spatial information about a particular lipid is completely lost. MSI means the recording of spatially-resolved mass spectra (normally from thin tissue slices) subsequent to the homogeneous application of a suitable MALDI matrix. This is actually the most serious problem regarding MSI, since the matrix application is not that easy. Afterwards, the peak intensities (in dependence on the position) were converted into gray or a color scale, to monitor the distribution of selected lipids.

Since no previous separation into the individual lipid classes is possible prior to the acquisition of the MALDI images, the suppression problems are quite similar to the MALDI MS analysis of lipid mixtures. Ion suppression effects affect the positive ion, as well as the negative ion mode. In the positive ion mode PC species are primarily detected [145], while the negative ion mode is normally dominated by phosphorylated and, particularly, sulfated (the sulfate residue represents a strong electrolyte) lipid species [126]. To these authors’ best knowledge, DHB and 9-AA are widely used in MSI [146]. Due to very recent reviews, these matrices, with respect to the MSI will not be discussed here. Instead, some selected applications of particular matrix compounds will be discussed.

It was recently reported that 1,5-DAN [147] has excellent properties (compared to the application of 9-AA) regarding the detection of PL in tissue slices. It was also shown that 1,5-DAN application yields a more representative MS profile of a natural PL mixture than 9-AA. It is important to note that 1,5-DAN may be applied by either spraying or sublimation, at which the latter usually gives a more homogeneous matrix coating [123]. Other matrices, namely, *N*-phenyl-2-naphthylamine [148] and the fluorescence probe DPH (1,6-diphenyl-1,3,5-hexatriene) [37] have been described to represent suitable matrices to unravel lipid localization by MALDI imaging.

One of the organs frequently investigated by MSI, is the brain. On the one hand, the brain is rich in PL and can be, thus, easily investigated without sensitivity problems. On the other hand, there are many lipid-related diseases (such as Alzheimer’s disease, AD) which affect the composition of brain. For instance, it was shown that plaque-associated sphingolipid alterations may be indicative of AD and can be monitored by MSI [149]. Somewhat later, the same authors have extended their study to other lipid classes, such as PL, and lyso-PL (LPL), which were confirmed to be deposited at the same positions as the disease-related amyloid β peptide isoforms [150]. This study was performed at a high spatial resolution (10 μm), which is currently the maximum which can be achieved by the MALDI MS. The minimum laser spot size is determined by the required sensitivity because a reduced laser spot size results in a lower yield of ions. Finally, an altered glycolipid metabolism was also detected in the AD brain samples [151]. Furthermore, MALDI imaging was used to elucidate alterations in the lipid architecture of brains from patients with Huntington disease [152].

In addition to the brain, MALDI imaging is used in cancer research [153,154]. It has been shown that healthy and cancerous tissues can be differentiated by differences in the lipid architecture which is evident from the images. However, further methodological improvements regarding specificity and sensitivity are still required [155].

## 4. Summary

MALDI-TOF MS is commonly used to analyze polar water-soluble molecules (such as proteins and peptides), although apolar molecules, such as lipids can be analyzed, too. Lipid analysis by the MALDI-TOF MS has gained particular interest since MSI is increasingly used to investigate the (phospho)lipid distributions in biological tissues. In addition to parameters, such as the solvent system, the laser fluence, the type of the mass spectrometer and the mass analyzer used, the matrix itself also plays a pivotal role. It must be explicitly noted that some lipid classes (such as TAG) are not detectable at all if the “wrong” matrix is used. In particular, to overcome ion suppression effects and to avoid a time-consuming sample preparation, a careful selection of the most suitable matrix is necessary. Although there were (and are) many studies where the authors tried to rationally design optimized MALDI matrices, many of these attempts were not really successful—the search for the most suitable matrix for a dedicated analyte is still an empirical task and many, so far widely used matrix compounds, were found by accident. It would be a major progress if the optimum matrix could be calculated by quantum chemical methods, since this would make many experimental studies unnecessary. For the detection of promising biomarkers in tissues related to distinct lipid-related diseases, suitable matrices for the MALDI imaging need to be investigated, continuously, to get the chance to obtain unequivocal diagnostic results.

## Figures and Tables

**Figure 1 biomolecules-08-00173-f001:**
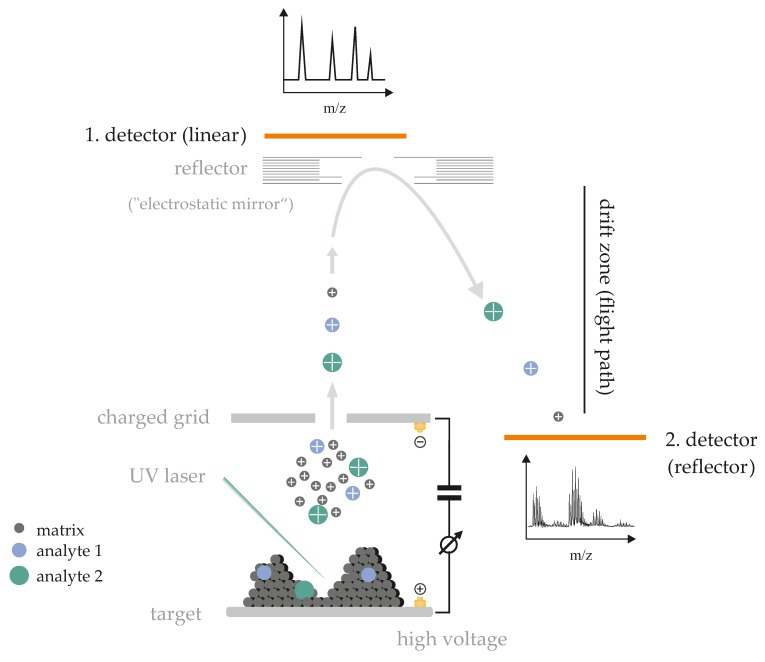
Simplified schema of the positive ionization matrix-assisted laser desorption/ionization-time-of-flight (MALDI-TOF) process occurring in the mass spectrometer (for details see text and Reference [18]). The influence of the detection using either the linear or the reflector mode is emphasized in the figure.

**Figure 2 biomolecules-08-00173-f002:**
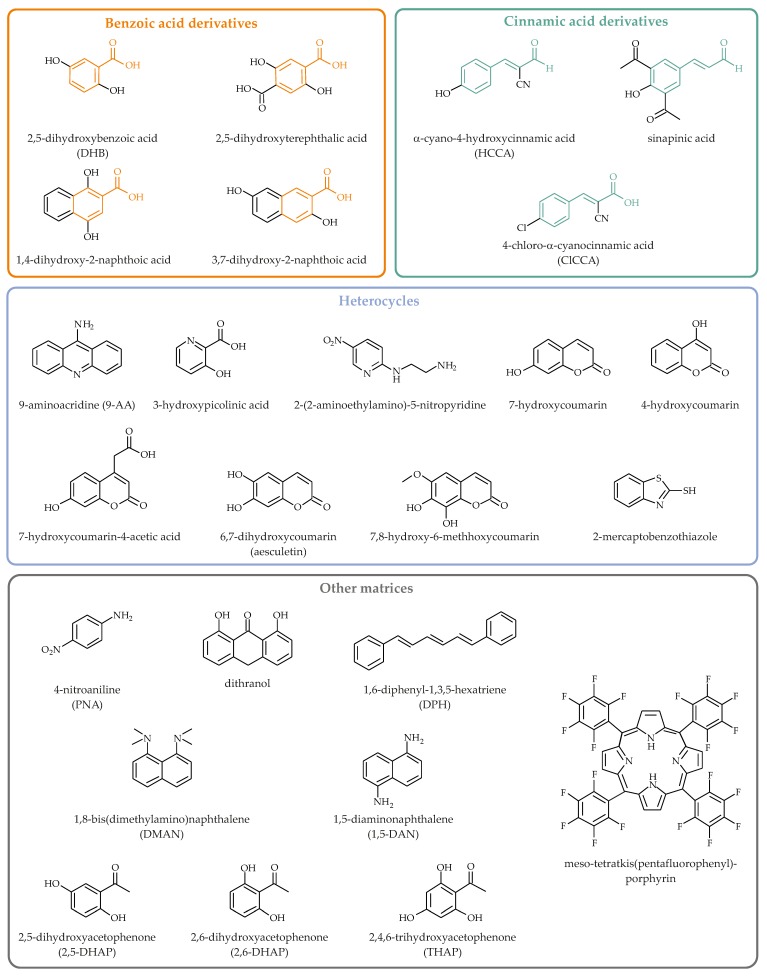
Chemical structures of the organic MALDI matrices which will be explicitly mentioned in this review. The backbones of both benzoic acid (orange) and cinnamic acid (green) derivatives are highlighted. Please note that this is only a small selection and many other compounds were also found to be useful MALDI matrices.

**Figure 3 biomolecules-08-00173-f003:**
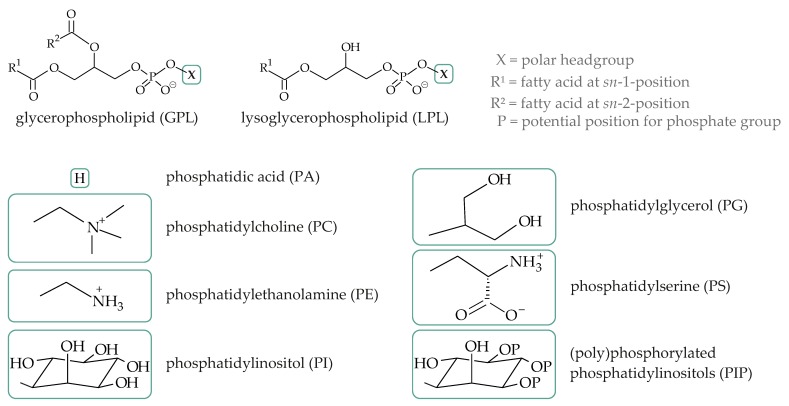
Chemical structures of selected phospholipid (PL) classes. Only the characteristic PL headgroups are shown. “R^1^” and “R^2^” denote different acyl (or in selected cases also alkyl) chains. Please note that the majority of physiologically-relevant lipids possess a saturated and an unsaturated fatty acyl moiety at the *sn-1* and the *sn-2* position, respectively. Although there also are alkyl-acyl and alkenyl-acyl PL, only the ester-linked compounds are shown here.

**Figure 4 biomolecules-08-00173-f004:**
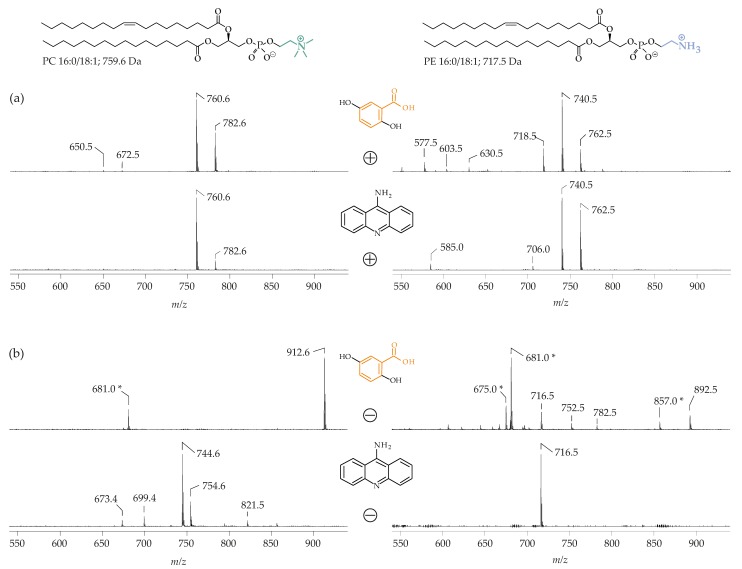
(**a**) Positive ion MALDI-TOF mass spectra of 1-palmitoyl-2-oleoyl-*sn*-phosphatidylcholine (PC 16:0/18:1; M = 759.6 g/mol, (**left**)) and 1-palmitoyl-2-oleoyl-*sn*-phosphatidylethanolamine (PE 16:0/18:1; M = 717.5 g/mol; (**right**)); (**b**) Negative ion MALDI-TOF mass spectra of PC 16:0/18:1 (**left**) and PE 16:0/18:1 (**right**). In all cases, the samples (0.2 mg/mL) were diluted 1:1 (*v*/*v*) either with a 2,5-dihydroxybenzoic acid (DHB; 0.5 M in methanol) or 9-aminoacridine (9-AA; 10 mg/mL in isopropanol/acetonitrile; 6/4, *v*/*v*). All peaks are marked according to their *m*/*z* ratio. DHB matrix peaks are marked with asterisks. This figure was reprinted with permission from [96] and slightly modified.

**Figure 5 biomolecules-08-00173-f005:**
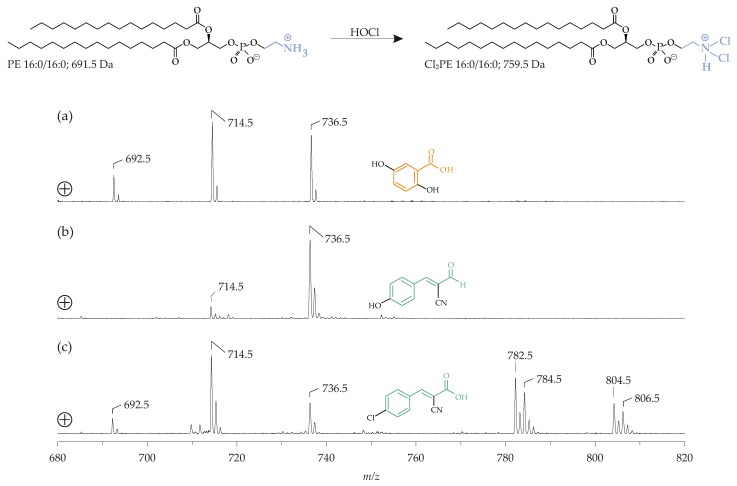
The MALDI-TOF mass spectra (positive ionization mode) of 1,2-dipalmitoyl-*sn-*phosphatidylethanolamine (PE 16:0/16:0; M = 691.5 g/mol), after treatment with a ten-fold molar excess of hypochlorous acid (HOCl) leading to the dichloramine of PE (Cl_2_PE) 16:0/16:0 using the following matrices: (**a**) DHB; (**b**) α-cyano-4-hydroxycinnamic acid (HCCA), and (**c**) 4-chloro-α-cyanocinnamic acid (ClCCA). Please note the exclusive detectability of the Cl_2_PE 16:0/16:0 as singly- and doubly-sodiated adducts at *m*/*z* 782.5 and 804.5, respectively, in the case of ClCCA (**c**). In all other cases, only PE 16:0/16:0 could be detected as protonated (*m*/*z* 692.5), sodiated (*m*/*z* 714.5), and as [(PE 16:0/16:0) − H + 2Na]^+^ at *m*/*z* 736.5. Reprinted with permission from Elsevier (with modifications) [104].

**Figure 6 biomolecules-08-00173-f006:**
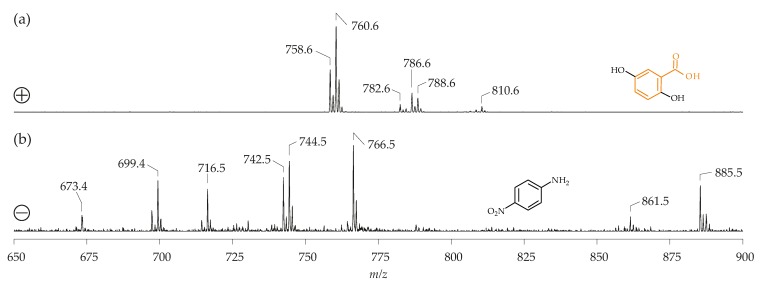
(**a**) MALDI-TOF mass spectrum (positive ionization mode) of an organic hen egg yolk extract recorded in the presence of DHB; (**b**) MALDI-TOF mass spectrum (negative ionization mode) of an organic hen egg yolk extract using 4-nitroaniline (PNA). Both spectra were recorded under exactly the same conditions and in the same solvents. More detailed assignments are available in (reprinted and modified with permission) [136].

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
