# Peer review of "Recent Developments of Useful MALDI Matrices for the Mass Spectrometric Characterization of Lipids"

_biomolecules, 2018, doi:10.3390/biom8040173_

Round 1
Reviewer 1 Report
The review, Leopold et al provide a concise overview on matrix developments for MALDI based lipid analysis in biological matrices.
The review is well written and covers major aspects of the field and is certainly of interest to the community. However, in its current form the manuscripts falls a bit short in covering some major recent developments in MALDI imaging MS.
This includes three aspects that should be included:
1) The use of 1,5 DAN for lipid imaging either by sublimation or dry spraying.
2) The compatibility of 1,5 DAN sublimation for subsequent IHC and other IMS modalities. (Kaya 2017 Anal Chem PMID: 28318232; Kaya et al 2017 ACS Chem Neurosci PMID: 28925253, Michno 2018 Anal Chem, PMID: 29856605)
3) Detailed applications of 1,5 DAN for e.g. probing Alzheimer Disease Pathology (Kaya, Brinet ACS Chem Neurosci 2017 PMID: 27984697 and 2018 PMID: 29648443;, Michno et al 2018 BBA Proteins/Proteomics PMID: 30273679)
Minor Comments:
Reference 15: Please update the reference
Reference 12, 16, 22, 79. Please include current literature and not book chapters.
Reference 25: Please include Caprioli and Murphy in Chem Rev PMID: 21942646
Author Response
See the attached letter file, please

Reviewer 2 Report
This article appears to draw heavily from Fuchs and Schiller, Current Organic Chemistry, 2009, 13, 1664-1681 ("Recent Developments of Useful MALDI Matricies for the Mass Spectrometric Characterisation of Apolar Compounds"). Running the submitted manuscript through TurnItIn (anti-plagiarism software) shows many sections in the main text that are nearly copied verbatim, with only one or two words changed to prevent outright plagiarism. My recommendation is to reject, or only accept after the authors have made significant changes to the text to remove the apparent similarities in the text.
Below are some suggestions and typographic/grammatical corrections for the authors to consider with their revisions:
line 71: It is probably worth explicitly stating either nitrogen laser (337nm) or Nd:YAG laser (355nm) at this stage. It would also be useful to mention what advantage the longer wavelength of the Nd:YAG laser affords. I believe that this is reduced fragmentation due to the lower photon energy.
line 90: remove the word "processes"
line 109: "try and error" should be "trial and error"
line 153: why is the drop and dry method not useful for quantitative analysis? I assume that this is because it does not provide a homogenous matrix deposition, but the authors should explicitly state this.
line 163: " a handful is..." should be "a handful are..."
line 217: remove 'the' in "Another method for the FFA..."
line 238: Can the authors comment on the utility of MALDI for analysing fatty acids that have been functionalised with a charge switch derivative, such as AMPP which is used for LCMS analysis?
line 251: remove the word "anymore" at the end of the sentence.
line 324: I would probably use the word hinders rather than aggravates in this sentence
line 528: it is unclear if the authors are still referring to DHB in this sentence
Author Response
See the attached letter file, please

Round 2
Reviewer 2 Report
The authors have made significant efforts to modify the text of the manuscript and should be commended. I feel that the revised text and added sections have strengthened the manuscript overall and it should now be accepted for publication after the below suggestions have been considered.
Please note that I have limited my comments and corrections to the modified or added sections of the manuscript.
Line 53: insert comma after the word "respectively"
Line 54: remove "the" before MALDI after the colon
Line 55: "ESI proceeds at ambient conditions" would sound better as "the ESI process occurs under ambient conditions"
Line 74: Change "...N2 lasers have a lower lifetime..." to "...N2 lasers usually have a shorter lifetime..."
Line 74/75: It is important to point out to the reader that it is the photon energy that is lower (as opposed to the pulse energy). The end of this sentence would read better as "moreover, the photon energy at 355nm is lower than 337nm."
Line 140: "The matrix should be stable under high vacuum conditions FOR as long as possible" insert the word FOR
Line 251: This makes it sound like BednaÅ™Ãk et al. (2018) are also performing ozone induced dissociation in their study, when they are definitely not. It would be advisable to include a reference to Paine et al. [Angewandte Chemie, 57, p10530 (2018)] after the word ozone, since this is a MALDI imaging study that employs ozone.
Line 254: Along this line, there was a paper within the last year [Frankefater et al, Journal of the American Society for Mass Spectrometry, 29, p1688 (2018)] looking at these fatty acid derivatives by MALDI TOF.
Line 548: The end 'd' is missing from the word 'deprotonated'
Line 629: It looks like the Caprioli reference here has not been picked up by Endnote.
Line 634: It would be worth noting that the lipid extracts are homogenised
Author Response
See the attached letter file, please
